# Domestication and lowland adaptation of coastal preceramic maize from Paredones, Peru

Miguel Vallebueno-Estrada[1,2†], Guillermo G Hernández-Robles[2], Eduardo González-Orozco[1], Ivan Lopez-Valdivia[1,2], Teresa Rosales Tham[3], Víctor Vásquez Sánchez[4], Kelly Swarts[5†], Tom D Dillehay[6,7], Jean-Philippe Vielle-Calzada[1]*, Rafael Montiel[2]*

[1]Grupo de Desarrollo Reproductivo y Apomixis, Unidad de Genómica Avanzada, Laboratorio Nacional de Genómica para la Biodiversidad, CINVESTAV, Irapuato, Mexico; [2]Grupo de Interacción Núcleo-Mitocondrial y Paleogenómica, Unidad de Genómica Avanzada, Laboratorio Nacional de Genómica para la Biodiversidad, CINVESTAV, Irapuato, Mexico; [3]Departamento de Antropología, Universidad Nacional de Trujillo, Perú, Trujillo, Peru; [4]Centro de Investigaciones Arquebiológicas y Paleoecológicas Andinas ARQUEBIOS, Trujillo, Peru; [5]Max Planck Institute for Developmental Biology, Tübingen, Germany; [6]Department of Anthropology, Vanderbilt University, Nashville, United States; [7]Escuela de Arqueología, Universidad Austral de Chile, Puerto Montt, Chile

*For correspondence:
vielle@cinvestav.mx (J-PV-C);
rafael.montiel@cinvestav.mx
(RM)

Present address: †Gregor
Mendel Institute, Austrian
Academy of Sciences, Vienna,
Austria

Competing interest: The authors
declare that no competing
interests exist.

Reviewing Editor: Detlef Weigel,
Max Planck Institute for Biology
Tübingen, Germany

**Abstract** Archaeological cobs from Paredones and Huaca Prieta (Peru) represent some of the oldest maize known to date, yet they present relevant phenotypic traits corresponding to domesticated maize. This contrasts with the earliest Mexican macro-specimens from Guila Naquitz and San Marcos, which are phenotypically intermediate for these traits, even though they date more recently in time. To gain insights into the origins of ancient Peruvian maize, we sequenced DNA from three Paredones specimens dating ~6700–5000 calibrated years before present (BP), conducting comparative analyses with two teosinte subspecies (*Zea mays* ssp. *mexicana* and *parviglumis*) and extant maize, that include highland and lowland landraces from Mesoamerica and South America. We show that Paredones maize originated from the same domestication event as Mexican maize and was domesticated by ~6700 BP, implying rapid dispersal followed by improvement. Paredones maize shows no relevant gene flow from *mexicana*, smaller than that observed in teosinte *parviglumis*. Thus, Paredones samples represent the only maize without confounding *mexicana* variation found to date. It also harbors significantly fewer alleles previously found to be adaptive to highlands, but not of alleles adaptive to lowlands, supporting a lowland migration route. Our overall results imply that Paredones maize originated in Mesoamerica, arrived in Peru without *mexicana* introgression through a rapid lowland migration route, and underwent improvements in both Mesoamerica and South America.

## Editor's evaluation

In this important article, the authors characterize ancient DNA from maize unearthed in archaeological contexts from Paredones and Huaca Prieta in the Chicama river valley of Peru, recovered by painstakingly controlled excavation. The genetic evidence, while from a small number of samples, is compelling, although the dating evidence has to rely on archaeological context, which fortunately is excellent. The difficulties of direct radiocarbon dating of the samples in this case are appropriately discussed by the authors.

## Introduction

Maize constituted 12% of global crop production in 2019, second only to sugar cane (*Food and Agriculture Organization of the United Nations, 2021*). Like many crop plants, global maize production is threatened by climate change, especially in the middle to low latitudes (*Li et al., 2022*) where maize is dominant. Maize has the allelic diversity to adapt, but much of this variation is partitioned differentially across populations (*Hufford et al., 2012b*; *Romay et al., 2013*; *Zila et al., 2013*). Understanding the development of population dynamics in maize not only allows better understanding of the evolutionary processes that produced a globally important crop but will highlight populations that can be used in breeding to adapt to changing climates.

Although the origins of maize (*Zea mays* ssp. *mays L.*) based on archeological data puzzled the scientific community for several decades (*Beadle, 1939*; *Mangelsdorf, 1974*; *Merrill et al., 1940*), the integration of genomic, archeological, and botanical evidence has identified the Balsas basin in central Mexico as the only center of origin for maize (*Iltis, 1983*; *Matsuoka et al., 2002*; *Ranere et al., 2009*), and that the divergence from its wild ancestor, *Zea mays* ssp. *parviglumis* (hereafter *parviglumis*), occurred about 9000 years ago (*Matsuoka et al., 2002*). Domestication occurred in a single event creating a monophyletic clade that includes all domesticated maize landraces and diverges from both *parviglumis* and *Zea mays* ssp. *mexicana* (hereafter *mexicana*) populations (*Matsuoka et al., 2002*). Genomic investigations of archeological samples from the Tehuacan highland site suggested that the dispersal of maize to the highlands of México was complex, as early-arriving maize populations retained higher levels of genomic diversity than expected for domesticated plants (*Ramos-Madrigal et al., 2016*; *Vallebueno-Estrada et al., 2016*). The constant gene flow between domesticated maize with already divergent populations of *parviglumis* and *mexicana* has contributed to the adaptation of maize to new environments and remains embedded in the genetic structure of its populations (*Swarts et al., 2017*; *van Heerwaarden et al., 2011*). Geographic areas of contact have been stable over time, as these teosinte populations have maintained a discrete distribution in central Mexico since the last glacial maximum (*Hufford et al., 2012a*). Gene flow from a sympatric *mexicana* population to domesticated maize populations has been associated with an altitudinal cline in the highlands of Mexico and Guatemala (*Hufford et al., 2013*; *van Heerwaarden et al., 2011*; *Wang et al., 2017*), and the genetic introgression from *mexicana* in the form of distinct chromosomal inversions has been associated with adaptation of maize to central Mexican highlands (*Hufford et al., 2013*; *Wang et al., 2017*). The genetic contribution of teosinte *mexicana* to Mexican highland landraces is about 20% (*Hufford et al., 2013*; *van Heerwaarden et al., 2011*), with an estimated time for this introgression of around 1000 generations (*Calfee et al., 2021*).

Archaeological evidence supports the dispersal of maize populations to South America to be associated with a Pacific lowland coastal corridor (*Bonavia and Grobman, 2017*; *Dillehay et al., 2008*; *Randolph, 1959*). Population substructure and differentiation patterns suggested independent adaptations to highland environments in Mesoamerica and South America; meanwhile, minimal population sub-structuring was detected between the lowlands of Mesoamerica and South America (*Swarts et al., 2017*; *Takuno et al., 2015*) indicating continuous gene flow over time. While the introgression from *mexicana* of chromosomal inversions on chromosomes 3, 4, and 6 has been shown to contribute to adaptation to Mesoamerican highlands (*Romero Navarro et al., 2017*), those regions were not detected in highland maize populations of South America (*Wang et al., 2017*) or North America (*Swarts et al., 2017*) which were also isolated from direct gene flow with *parviglumis* (*Kistler et al., 2018*). Although these inversions are specific to Mexican landraces, many maize populations across the Americas including South America show genome-wide admixture with *mexicana* (*Swarts et al., 2017*). Highland South American landraces also show phenotypic diversity relative to lowlands (*Bonavia, 2013*), as well as specific cytogenetic characteristics such as the absence of supernumerary highly heterochromatic B chromosomes in Peruvian landraces, resulting in the so-called Andean Complex (*McClintock et al., 1981*). These unique characteristics have puzzled the scientific community regarding the origin and adaptation of the Andean Complexes (*Bonavia, 2013*; *Goodman and Bird, 1977*; *Grobman, 1961*; *McClintock et al., 1981*; *Randolph, 1959*; *Wilkes, 1979*).

The archeological expeditions at the sites of Paredones and Huaca Prieta in the coastal desert of north Peru yielded a robust collection of ancient maize remains that provide a unique opportunity to investigate the chronology, landrace evolution, and cultural context associated with early maize dispersal in South America (*Bonavia and Grobman, 2017*; *Dillehay et al., 2012*; *Grobman et al.,*

**eLife digest** The plant we know today as maize or corn began its story 9,000 years ago in modern-day Mexico, when farmers of the Balsas River basin started to carefully breed its ancestor, the wild grass teosinte *parviglumis*. Recent discoveries suggest the crop may have started to travel to South America before its domestication was fully complete, leading to a complex history of semi-tamed lineages evolving in parallel in different regions. For example, 5,300-year-old corn specimens found in Tehuacán, in central Mexico, still genetically and morphologically resemble teosinte. Meanwhile, cobs harvested about 6,700 to 5,000 years ago on the northern coast of Peru – 3800km away from where maize was first domesticated – look like the ones we know today.

Vallebueno-Estrada et al. aimed to explore the evolutionary history of this Peruvian maize, which was discovered at the archaeological coastal site of Paredones. To do so, they extracted and sequenced its genetic information, and compared these sequences with those from modern varieties of lowland and highland maize, as well as from teosinte *parviglumis* and teosinte *mexicana*.

The analyses showed that the ancestor of the Paredones maize emerged from teosinte *parviglumis* like any other lineage, but that it was already domesticated when it started to spread South; by the time it was present in Peru 6,700 years ago, it was genetically closer to modern-day crops. This early departure is consistent with the fact that the Paredones specimens lacked teosinte *mexicana* genetic variants; this highland relative of lowland *parviglumis* is believed to have interbred with maize lineages from Central America more recently, when these were brought to higher altitudes.

The presence of genetic marks tailored to low-elevation regions suggested that the Paredones maize lineage migrated through a coastal corridor connecting Central and South America, arriving in northern Peru about 2,500 years after first arising from teosinte *parviglumis* in Central America around 9,000 years ago. Under the care of rapidly developing Central Andean societies, the crop then evolved to adapt to its local conditions.

Maize today has spread to all continents besides Antarctica; we produce more of it than wheat, rice or any other grain. How our modern varieties will adapt to the environmental constraints brought by climate change remains unclear. By peering into the history of maize, Vallebueno-Estrada et al. hope to find genetic variations which could inform new breeding strategies that improve the future of this crop.

*2012*). These findings include one charred cob fragment dated 6775–6504 calibrated BP (at 95.4% probability), and other burned cobs stratigraphically dated to similar and later ages (*Bonavia and Grobman, 2017*; *Dillehay et al., 2012*; *Grobman et al., 2012*), representing the most ancient maize macro-specimens found to date. Strikingly, and in contrast to Mexican cob fragments from Guilá Naquitz dating approximately the same time or slightly younger (6235 calibrated BP) (*Benz, 2001*), the Paredones and Huaca Prieta specimens are robust, slender and cylindrical 2.4–3.1 cm long cobs, with eight rows of kernels consistent with the hypothetical Proto-Confite Morocho landrace (*Benz, 2001*; *Dillehay et al., 2012*; *Grobman et al., 2012*). Recently, it has been proposed that the first maize lineages arriving in South America were partial domesticates, locally evolving the full set of domestication traits due to reduced gene flow from wild relatives that enhanced anthropogenic pressures (*Kistler et al., 2020*; *Kistler et al., 2018*). However, it is not clear how fast this process could have been and if the earliest archeological samples found in South America were partially or fully domesticated. In addition, the expectation on the phenotype of those hypothetical samples is not clear.

Here we present the genomic analysis of three ancient specimens belonging to the earliest cultural phase of Paredones and radiocarbon dating 6775–6504, ~5800 to 5400 (dated by direct association with wood charcoal), and 5583–5324 $2\sigma$ calibrated years BP at 95.4% probability. To reveal the population context of their origin and domestication, we conducted comparisons with *parviglumis*, *mexicana* and extant maize landraces. Also, to explore if these ancient maize samples exhibit some evidence of *mexicana* gene flow, we performed D-statistics under several experimental designs, comparing them to extant maize and *parviglumis* populations. Finally, we did a comparison with previously published data from extant highland and lowland Mesoamerican and South American landraces, to identify signatures of specific adaptation that could bring insights into the specific improvements that this

maize went through in both Mesoamerica and South America. Our results provide evidence that ancient Peruvian maize originated in Mesoamerica as all landraces found to date, followed by a rapid dispersal into the lowlands of South America, and subsequently was subjected to local adaptation processes.

## Results

### Paredones ancient maize sampling

The maize macroremains were collected as part of published excavations at the Paredones and Huaca Prieta sites (*Grobman et al., 2012*). Macroremains from both sites were excavated in deeply stratified and undisturbed cultural floors. Stratigraphic Units 20 and 22 at Paredones are the archeological component with the largest and most diversified amount of maize remains, with the oldest $^{14}$C dated cobs. The oldest cobs derive from near the base of this unit, in a single, discrete and intact floor of ~2 cm in thickness and at 5.5 m in depth from the present-day surface (*Bonavia and Grobman, 2017*; *Grobman et al., 2012*). The dated remains at both sites are chrono-stratigraphically bracketed by and in agreement with more than 165 dates from mound and off-mound contexts that were obtained by Accelerator Mass Spectrometry (AMS) and Optically Stimulated Luminescence (OSL) (*Bonavia and Grobman, 2017*; *Dillehay, 2017*; *Dillehay et al., 2022*). No taphonomic or other disturbing cultural or geological features were observed in any excavation units that would have altered the integrity and intactness of strata containing the maize remains (Material and Methods). All radiocarbon-dated remains were assayed by the SHCal04 Southern Hemisphere Calibration 0–11.0 calibrated kyr BP curve (*McCormac et al., 2004*).

In 2020, Dillehay and geologists Steven Goodbred and Elizabeth Chamberlain carried out excavations in a Preceramic domestic site (S-18) located ~3.2 km north of Huaca Prieta. Preceramic corn remains were encountered consistently in hearths and in the upper to lower intact cultural layers of the site. As with parts of the Paredones and Huaca Prieta sites, the lower hearths and strata contained uncharred cobs 2.6–3.1 cm long, slender and cylindrical with eight rows of kernels, of the smaller and earliest type of identified corn species Proto Confite Morocho (*Grobman et al., 2012*). The middle to upper strata yielded the known later and slightly larger Preceramic varieties of Confite Chavinese and Proto Alazan. An OSL date from a discrete and intact lower layer containing a hearth with two unburned cob fragments of the Proto Confite Morocho variety assayed ~7000 +/- 630 years ago or 7560–6300 BP (*Chamberlain, 2019*). Wood charcoal from the hearth was processed at 7162–6914 calibrated BP (at 95.4% probability, AA75398), suggesting that the associated cob fragments date ~6800 years ago. In South America, maize micro remains (e.g. starch grains, pollen, phytoliths) have been estimated to date 7200–7000 calibrated BP (*Piperno, 2006*; *Piperno and Pearsall, 1998*; *Zarrillo et al., 2008*) at sites in southwest coastal Ecuador, located ~450 km north of Huaca Prieta and Paredones, and in other localities across the continent at ~6500 calibrated BP and later (*Kistler et al., 2018*).

Excavation Units 20 and 22 from Paredones are illustrated in *Figure 1A and B*. Three of the recovered maize samples (Par_N1, Par_N9, and Par_N16) are well-structured maize cobs deprived of seeds and showing morphological similarities to extant landraces. Par_N1 is the most ancient specimen, a burned maize husk and shank fragment dating 5900 ± 40 $^{14}$C years BP (6770–6504 calibrated BP at 95.4% probability, OS860020), and obtained from archeological Unit 22 (*Figure 1C* and *Figure 1—figure supplement 1*). The other two samples were found in Unit 20, Par_N9, a slightly charred maize husk fragment, that radiocarbon assayed at 5582–5321 calibrated BP (at 95.4% probability, AA86932), and Par_N16, an unburned cob fragment, which stratigraphically dated by direct association with wood charcoal in a hearth at 5603–5333 calibrated BP (at 95.4% probability, AA86937; see *Figure 1C* and *Table 1*). Other charred cobs from overlying, younger strata in these units or from other units assayed between 4800–3800 calibrated BP (at 95.4% probability). These dates are stratigraphically bracketed by radiocarbon assayed intact hearths and prepared floors that are in complete chrono-stratigraphic agreement (*Bonavia and Grobman, 2017*; *Dillehay et al., 2012*; *Grobman et al., 2012*). Par_N1 is older than any other maize macro-specimen found to date (*Torres-Rodríguez et al., 2018*).

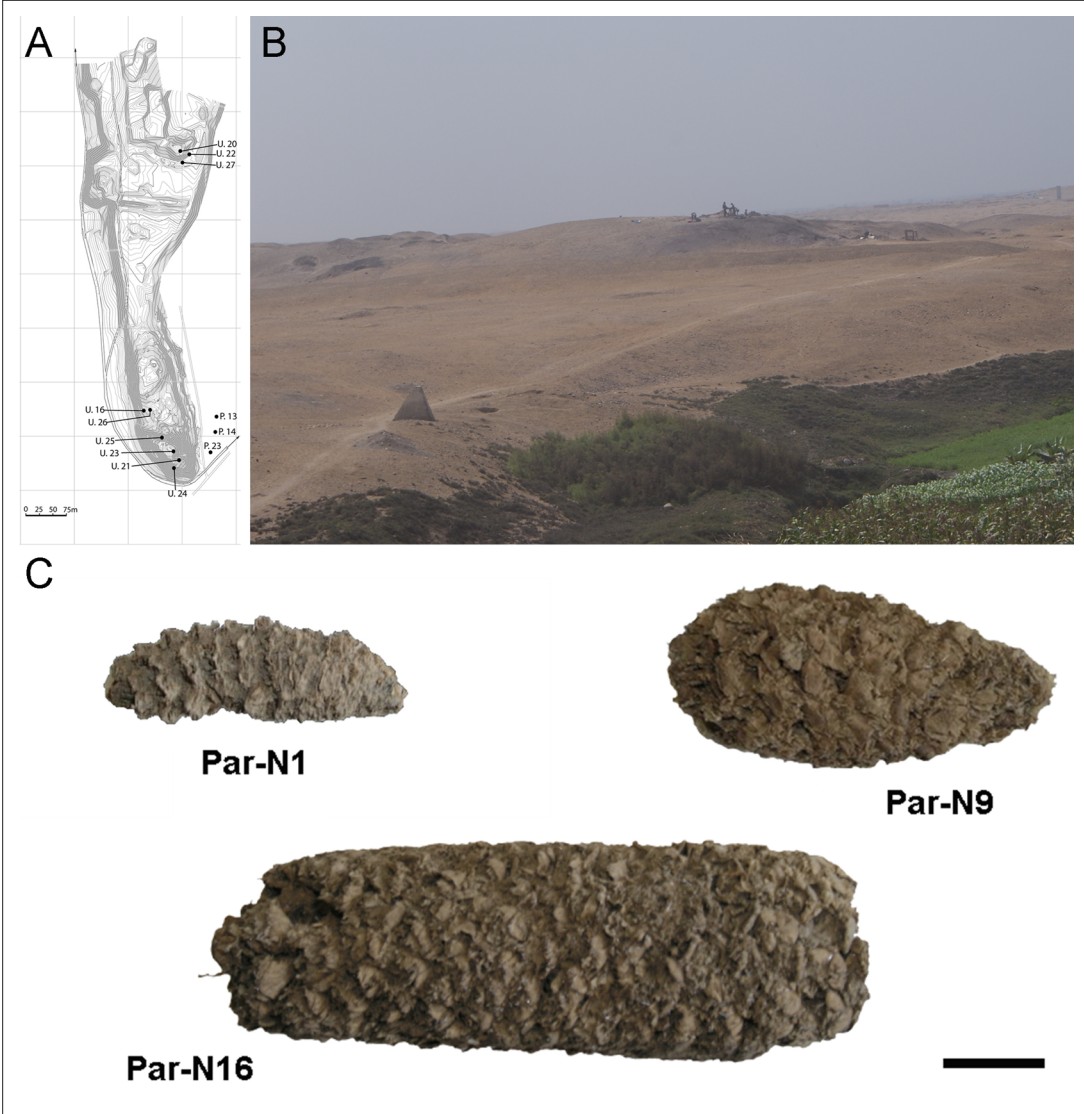

**Figure 1.** Archeological site and specimens of Paredones. (**A**) Topographic contour map of Huaca Prieta and Paredones (units U20, U22, and U27) coastal sites, showing the location excavation units. (**B**) The Paredones mound during archeological excavations. (**C**) Maize specimens Par_N1 (dating 6775–6504 calibrated years BP), Par_N9 (dating to 5800–5400 calibrated years BP), and ParN16 (dating 5583–5324 calibrated years BP); Scale bar = 1 cm.

The online version of this article includes the following figure supplement(s) for figure 1:

**Figure supplement 1.** Intact stratigraphy of thin floors in Unit 22.

## Paleogenomic characterization of ancient maize samples

To determine the genomic constitution and degree of genetic variability present in the 6775–5324 calibrated years BP (at 95.4% probability) maize of Paredones, we extracted DNA and conducted whole-genome shotgun sequencing in specimens Par_N1, Par_N9, and Par_N16. Since the endogenous DNA content of all three specimens was low (0.2% for Par_N1 and Par_N9; 1.1% for Par_N16), we conducted in-depth whole-genome shotgun sequencing of high-quality libraries under Illumina platforms, generating 622 million (M) quality-filtered reads for Par_N1, 423 M for Par_N9, and 392 M for Par_N16. Due to its higher endogenous DNA content (one order of magnitude larger), we further sequenced the Par_N16 library, obtaining 459 M additional reads, to generate a total of 851 M for this sample (*Table 2*). Comparison with version 3 of the B73 maize reference genome resulted in 1,320,284 (Par_N1), 1,034,544 (Par_N9), and 15,023,803 (Par_N16) reads mapping to either repetitive (33.4% for Par_N1; 34% for Par_N9; 34.8% for Par_N16) or unique (66.5% for Par_N1; 66% for Par_N9; 65.2% for Par_N16) genomic regions, for a total virtual length of 52.2 Mb (Par_N1), 40.8 Mb (Par_N9), and

**Table 1.** Radiocarbon and calibrated dates of maize specimens from Paredones.

| Lab no. | Site | Associated Dating no. | Unit / stratum | 14 $^c$ years BP | 95.4% probablity | δ13C value | Dated Material |
|---------|------|----------------------|----------------|------------------|------------------|------------|----------------|
| Par_N1 | Paredones | OS86020 | 22/18 | 5900±40 | 6775–6504 | −10.3 | Husk/shank fragment attached to cob |
| Par_N9 | Paredones | AA86932 | 20/6b-18 | 4770±35 | 5582–5321 | −23.5 | Husk fragment attached to cob* |
| Par_N16 | Paredones | AA86937 | 20/6b-18 | 4849±31 | 5603–5333 | −25.8 | Wood charcoal in associated hearth† |

*Aberrant δ13C assay. Attachment to cob and molecular data confirm maize.
†Maize cob directly associated with hearth.

471.65 Mb (Par_N16) of the unique maize genome (**Table 2**). Average mapping quality in Phred score was 31.9 for Par_N1, 32.1 for Par_N9 and 34 for Par_N16; this is reflected in the estimated error rate of 1.19E-02 for Par_N1, 9.59E-03 for Par_N9 and 1.05E-02 for Par_N16 (**Table 2**). Reads contained signatures of DNA damage typical of postmortem degradation in ancient samples, including overhangs of single-stranded DNA, 13–20% cytosine deamination and fragmentation due to depurination (**Dabney et al., 2013**) resulting in median fragment lengths of 36 bp for all three samples. A total of 42–53% of all covered sites had signatures of molecular damage (**Figure 2**). This damage pattern is an indication that this is ancient endogenous DNA and does not represent DNA contamination from extant sources. After mapping reads corresponding to unique genomic regions, Par_N1, Par_N9 and Par_N16 yielded approximately 16.9 M (Par_N1), 12.1 M (Par_N9), and 334.36 M (Par_N16) unique genomic sites spread across all 10 chromosomes at an average depth of 1.2 X (**Table 3** and **Figure 2— figure supplements 1–4**), which were used as a platform for subsequent studies.

When compared to the B73 reference genome, Par_N1, Par_N9, and Par_N16 yielded 21,123, 15,554, and 275,990 single nucleotide polymorphisms (SNPs), respectively. To eliminate any potential miscalls caused by postmortem damage, all SNPs corresponding to a possible cytosine (C) to thymine (T), or guanine (G) to adenine (A) transitions were not considered for subsequent analysis (molecular damage filter) (**Gilbert et al., 2003**; **Hofreiter et al., 2001**; **Table 4**). All SNPs corresponding to insertions or deletions (INDELs) were also eliminated. Using a previously reported pipeline (**Vallebueno-Estrada et al., 2016**), Par_N1, Par_N9, and Par_N16 yielded 2,886, 1,888, and 121,842 intersected positions with called genotypes (genotype calls) included in the HapMap3 maize diversity panel, most of which were only covered at 1 X due to the low amount of endogenous DNA recovered (**Table 4**). Despite this low coverage depth, the vast majority corresponded to a previously reported HapMap3

**Table 2.** Paleogenomic characterization of three ancient maize samples from Paredones.

| | Par_N1 | Par_N9 | Par_N16 |
|---|--------|--------|---------|
| Total number of raw reads | 623,686,255 | 423,856,284 | 851,330,235 |
| Total number of quality sequences | 622,438,882 | 423,472,877 | 850,326,750 |
| Number of sequences mapping to genome | 1,320,284 | 1,034,544 | 15,023,803 |
| Number of sequences mapping to repetitive regions | 441,442 | 351,883 | 5,228,275 |
| Number of sequences mapping to the unique genome | 878,842 | 682,661 | 9,795,586 |
| Total length (Mb) | 52.2 | 40.80 | 471.65 |
| Average read length (bp) | 59.41 | 59.90 | 48.15 |
| Total coverage (Mb) | 16.90 | 12.100 | 334.36 |
| Average quality (Phred) | 31.9 | 32.1 | 34 |
| Error rate (mismatches / bases mapped (cigar)) | 1.19E-02 | 9.59E-03 | 0.01059832 |

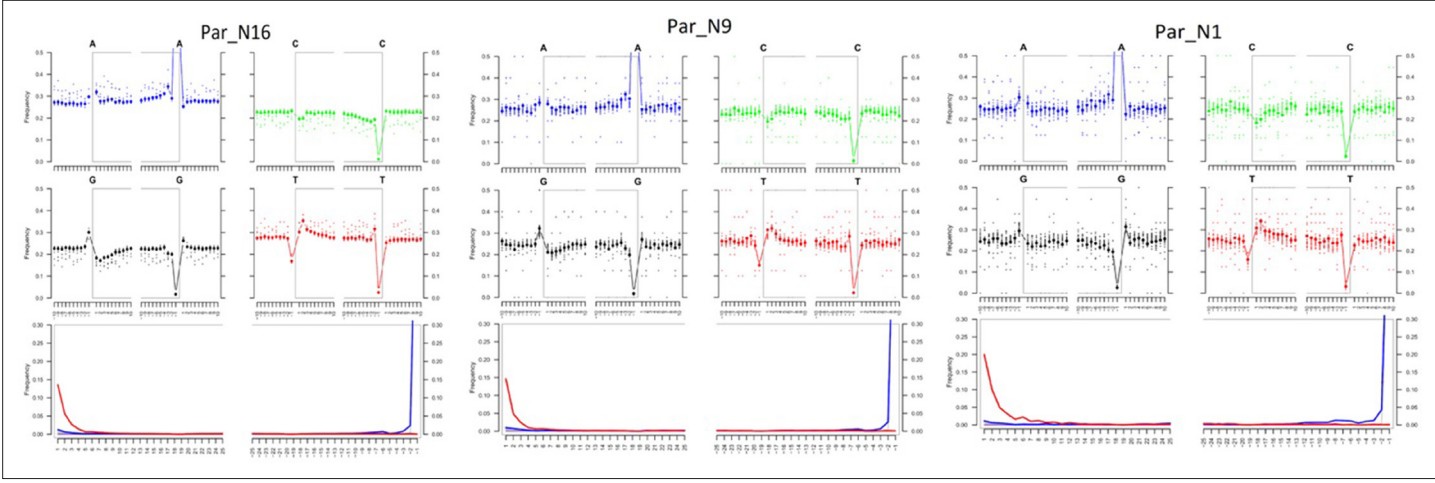

**Figure 2.** Post-mortem DNA damage and fragmentation patterns of ancient maize samples. DNA composition around read-termini (top four plots), and DNA mis-incorporation errors relative to the 5' and 3' read (bottom plot); the two distributions for post-mortem damage signatures (C to T and G to A) are shown in red and blue respectively, while other types of substitutions are shown in gray.

The online version of this article includes the following figure supplement(s) for figure 2:

**Figure supplement 1.** Mapped fragment length plots.

**Figure supplement 2.** Total coverage of the unique genome for the three ancient samples.

**Figure supplement 3.** Distribution of all genomic regions covered by reads from Par_N16.

**Figure supplement 4.** Genomic distribution of SNPs for the ancient samples.

**Figure supplement 5.** Distribution of genotype calls shared between Par_N16, Par_N1, and the HapMap3.

allele (98.8% for Par_N1, 98.9% for Par_N9, and 99.1% for Par_N16), suggesting that this dataset provides an accurate paleogenomic representation of maize that can be used to determine its evolutionary trend. To determine if the specific elimination of C to T and G to A modifications could bias the results in favor of maize rather than teosinte alleles, an additional database was generated in which all transitions were eliminated (i.e. only transversions were included) in Par_N16 only, because it was the only sample with enough sequencing data to conduct this experiment. This second database consisted of 64,118 transversions SNPs, intersected between this sample and the HapMap3 panel. With this database we conducted parallel analyses, which results are shown in the corresponding sections.

**Table 3.** Total number of unique genomic sites covered at variable depths in ancient Paredones samples.

| Depth | Par_N1 | Par_N9 | Par_N16 |
|---|---|---|---|
| 1 | 15,679,844 | 11,436,116 | 278,622,390 |
| 2 | 1,068,931 | 583,349 | 44,373,018 |
| 3 | 130,670 | 60,892 | 7,898,502 |
| 4 | 24,923 | 11,685 | 1,875,328 |
| 5 | 8,010 | 5291 | 644,804 |
| 6 | 3759 | 2309 | 299,729 |
| 7 | 2892 | 1189 | 167,142 |
| 8 | 1423 | 967 | 103,162 |
| 9 | 1026 | 837 | 68,272 |
| 10 | 869 | 774 | 47,003 |
| >10 | 10,419 | 9341 | 155,805 |

## Relationship between ancient maize, extant landraces, and Balsas teosinte

To better understand the origin and domestication of South American maize, we explored the evolutionary relationship between Paredones specimens, teosinte *parviglumis* and *mexicana*, and extant maize landraces. We inferred a bootstrapped maximum-likelihood (ML) tree topology through patterns of population divergence applied to genome-wide polymorphisms. Intersected positions among the three ancient Paredones samples were scarce (*Figure 2—figure supplement 5*); therefore, topologies were constructed individually for each DNA sample, based on the intersection of genotype calls between each of the samples and the maize HapMap3 dataset that includes B73 as a reference genome (including

**Table 4.** Number of SNPs and genotype calls recovered from ancient Paredones samples.

| | Par-N1 | Par-N9 | Par-N16 | | Par-N16 transversions |
|---|---|---|---|---|---|
| Total number of SNPs | 21,123 | 15,554 | 275,990 | Total number of SNPs | 275,990 |
| Transitions C->T | 7505 | 5766 | 41,811 | Transitions C<->T | 76,990 |
| Transitions G->A | 7527 | 5617 | 41,669 | Transitions G<->A | 76,909 |
| INDELS | 609 | 423 | 19,790 | INDELS | 19,790 |
| Quality SNPs | 5482 | 3748 | 192,510 | Quality SNPs | 102,302 |
| Genotype calls included in HapMap3 | 2886 | 1888 | 121842 | Genotype calls included in HapMap3 | 64,118 |

major and minor frequency alleles), 22 maize landraces (including several originating in Mexico), 15 teosinte *parviglumis* inbred lines, two accessions of teosinte *mexicana*, and a single accession of *Tripsacum dactyloides* acting as the outgroup (*Supplementary file 1*). In the case of Par_N16, the resulting tree shows all maize landraces and teosinte accessions separated into two distinct clades, all derived from *Tripsacum* as previously reported (*Hufford et al., 2012b*; *Matsuoka et al., 2002*). Par_N16 is in a clade that includes extant maize landraces, and this is for all 10,000 bootstrap replicates tested. Par_N16 is not basal in its clade but fits robustly with *Chullpi* (AYA 32) – the only extant Peruvian landrace included in the reference panel – in a derived position, closely clustering with South American landraces such as *Cravo Riogranense* (RGSVII) and *Araguito* (VEN 568). These relationships indicate that the ancient samples are monophyletic with modern maize, supporting a single domestication event, and that they are most closely related to modern samples from the same region, strongly suggesting an ancestral relationship between them and modern South American germplasm (*Figure 3* and *Figure 3—figure supplement 1*). In the case of Par_N1 and Par_N9, and although genotype calls intersected with HapMap3 were scarce (2886 and 1888, respectively), the resulting topology is equivalent, with both samples clustering at the same position as Par_N16 (*Figure 3—figure supplements 2 and 3*). As other ancient samples previously analyzed, Paredones samples tend to have long branches in phylogenies, which can be explained by isolation by time. On the other hand, the fact that 3 independent samples present the same position in the phylogeny indicates that molecular damage, which is random, is not driving their phylogenetic signal. A parallel analysis that only included transversions showed the same topology, where Par_N16 groups with the South American landraces within the maize monophyletic clade (*Figure 3—figure supplement 4*). This shows that the phylogenetic position of the Paredones ancient samples is not biased by the molecular damage filter. Thus, based on genome-wide relatedness, Paredones maize clusters with extant domesticated Andean landraces, supporting both, a single origin for maize and that these Peruvian samples were already domesticated by ~6700 BP.

## Tests of gene flow from mexicana

Par_N16 was the only sample with enough DNA sequence data to perform this analysis. All the samples showed the same phylogenetic position; therefore, Par N 16 was considered to be representative of ancient Paredones maize. To investigate the genetic relationship of this maize with teosinte *mexicana*, we estimated D-statistics in the form D(*parviglumis*, *mexicana*, TEST, *Tripsacum*) that test the hypothesis of Incomplete Lineage Sorting (ILS) due to persistence of polymorphisms across different divergence events, against an imbalanced gene flow over derived alleles from *parviglumis* to TEST, and *mexicana* to TEST (*Figure 4—figure supplement 1*). We used highland *Palomero Toluqueño* (PT2233) as a positive control, in the form D(*parviglumis*, *mexicana*, PT2233, *Tripsacum*), and lowland *Reventador* (BKN022) as a negative control, in the form D(*parviglumis*, *mexicana*, BKN022, *Tripsacum*). The results of multiple D-statistic distributions show that the positive control PT2233, with D>0, deviated from the balanced gene flow towards *mexicana*. Meanwhile, BKN022 remains in ILS balance, with D around 0. The D-statistics distribution of Par_N16 D(*parviglumis*, *mexicana*, ParN16, *Tripsacum*) is statistically similar to the distribution of *Reventador* (BKN022) (two-sample Kolmogorov-Smirnov, *P*=0.5814) and significantly different from the distribution of *Palomero Toluqueño* (two-sample Kolmogorov-Smirnov,

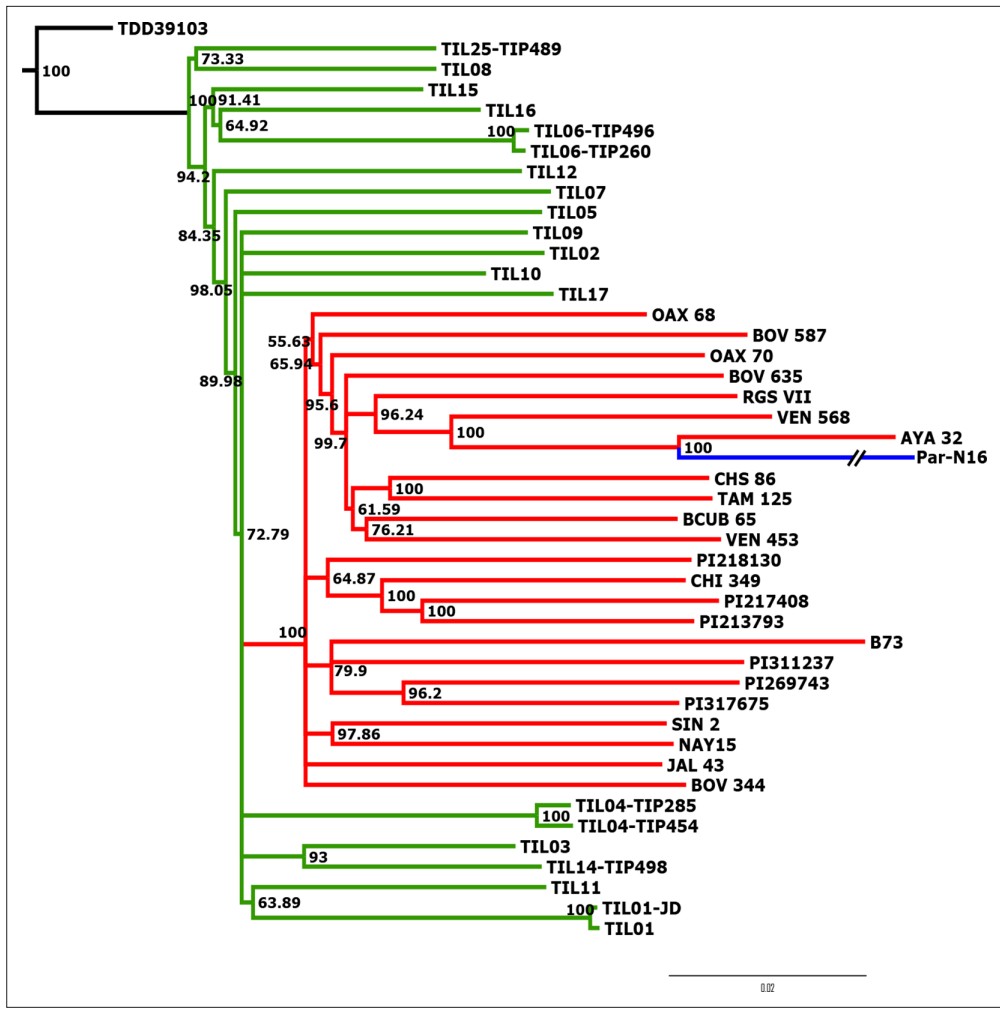

**Figure 3.** Advanced domestication of ancient Peruvian maize. Evolutionary relationships between ancient Par_N16 maize and its wild and cultivated relatives. ML tree from an alignment of 121,842 genome-wide genotype calls covering non-repetitive regions of the reference maize genome. The teosinte group is highlighted in green, the maize landrace group in red, and the ancient maize sample from Paredones in blue. The teosinte and landrace accessions follow previously reported nomenclatures and are described in the *Supplementary file 1*. The Par_N16 branch was cut for format reasons; a tree with the complete branch can be seen in *Figure 3—figure supplement 1*.

The online version of this article includes the following figure supplement(s) for figure 3:

**Figure supplement 1.** Evolutionary relationships between ancient Paredones maize Par_N16 and its wild or cultivated relatives.

**Figure supplement 2.** Evolutionary relationships between Paredones ancient maize Par_N9 and its wild or cultivated relatives.

**Figure supplement 3.** Evolutionary relationships between Par_N1 ancient maize and its wild or cultivated relatives.

**Figure supplement 4.** Evolutionary relationships between ancient Paredones maize Par_N16 and its wild or cultivated relatives in which only transversions were used.

p<0.0001) (*Figure 4—figure supplement 2*). The standard deviation of all 1000 jackknife replications is narrow in all cases (SD <0.001), suggesting that D values are consistent across the genomes. These results agree with the D statistics analysis in which only transversions were used (*Figure 4—figure supplement 3*), showing that the absence of significant gene flow between Par_N16 and *mexicana* is not biased by the molecular damage filter.

To further confirm the absence of *mexicana* introgression in Par_N16, we contrasted the gene flow between *mexicana* and *parviglumis* against the gene flow between *mexicana* and the test sample with D-statistics in the form D(TEST, *parviglumis*, *mexicana*, *Tripsacum*) (***Figure 4—figure supplement 4***). In this case, D<0 is an indication of a higher gene flow between *mexicana* and TEST than between *mexicana* and *parviglumis*. As expected, the highland control in the form D(PT2233, *parviglumis*, *mexicana*, *Tripsacum*) resulted in D<0, showing significantly higher gene flow with *mexicana* than the one observed between *mexicana* and *parviglumis*. Both the lowland negative control BKN022 and Par_N16 resulted in D>0, indicating low levels of gene flow between either BKN022 or Par_N16 and *mexicana* (***Figure 4***). In both cases, the narrow standard deviation of 1000 jackknife replications (SD <0.001) suggests that these D values are consistent across the genomes. The result of D>0 for the ancient Paredones maize and the fact that Par_N16 has a significantly higher D value than both lowland and highland controls (two-sample Kolmogorov-Smirnov, p<0.0001), was also confirmed with results from a parallel analysis conducted with transversions only (***Figure 4—figure supplement 5***), showing again that the lower degree of gene flow between *mexicana* and Par_N16 is not biased by the molecular damage filter. These results consistently show the absence of significant gene flow between Par_N16 and *mexicana*, implying that the lineage that gave rise to Paredones maize left Mesoamerica without relevant introgressions from this teosinte.

## Specific adaptation to lowlands in Mesoamerica and South America

Our phylogenetic analysis indicates that the lineage leading to Paredones maize left Mesoamerica already domesticated. However, it does not provide an assessment of how much adaptive variation derives from Mesoamerica (MA) and how much occurred after moving to South America (SA). To assess this, we identified in Par_N16 all covered SNPs with alleles previously reported to be adaptive to highlands and lowlands, specifically in Mesoamerica or South America by Takuno and coworkers (***Takuno et al., 2015***). These authors used genome-wide SNP data from 94 Mesoamerican and South American landraces and identified SNPs with significant $F_{ST}$ values to infer which allele was likely adaptive. For example, those SNPs showing significant $F_{ST}$ only in Mesoamerica, were characterized as adaptive for lowlands if they were at high frequency in the lowland population and at low frequency in the highland population, and vice versa. The same was applied for South America (***Takuno et al., 2015***). They identified 668 Mesoamerican and 390 South American adaptive SNPs, from which 32 and 20 were covered in Par_N16, respectively. In general, adaptive SNPs represented in Par_N16 were not clustered. The 20 South American adaptive SNPs are at a median distance of 8,301,843 bp, while the 32 Mesoamerican SNPs are at a median distance of 24,295,968 bp (***Supplementary file 2***). SNPs in five pairs from MA are closer than 100 bp between them, but each pair is at a considerable distance (beyond 1 cM) from each other and from other SNPs. This same happens for only one SNP pair from SA. Then, although at low proportions, the adaptive SNPs in Par_N16 are a *bona fide* representation of different genomic responses to selection pressures, and they are not significantly underrepresented (p=0.8009 and p=0.2962, for Mesoamerica and South America, respectively), relative to the coverage expectation of non-adaptive SNPs obtained from the same study (***Takuno et al., 2015***; see Materials and methods and ***Figure 5—figure supplement 1***). Also, there is no statistically significant difference in the proportion of adaptive SNPs covered in Par_N16 corresponding to Mesoamerica and South America (32/668 vs. 20/390, Fisher exact test, p=0.8832), suggesting an equivalent power of SNP detection in both regions.

We estimated the allelic similarity between Par_N16 and each test population (highland Mesoamerica, lowland Mesoamerica, highland South America, and lowland South America) in both adaptive and background SNPs. The allelic similarity is the average of the frequencies of the Par_N16 alleles in the intersected sites with each test population (see Materials and methods). Comparison of the similarity estimated from adaptive SNPs to similarities estimated from multiple random samples of background SNPs allows a quantification of the deviations from genome-wide similarity expectations. Each random sample contained the same number of SNPs as the number of adaptive SNPs covered in Par_N16 (32 for Mesoamerica and 20 for South America) (Material and Methods and ***Figure 5***). We analyzed some of these random samples and observed a similar behavior as the adaptive SNPs regarding the range of distances between SNPs (***Figure 5—figure supplement 2***). The mean genome-wide similarity of Par_N16 is 0.785 for highland and 0.800 for lowland populations in Mesoamerica, and 0.831 for highland and 0.812 for lowland populations in South America (***Figure 5***). Thus, at the

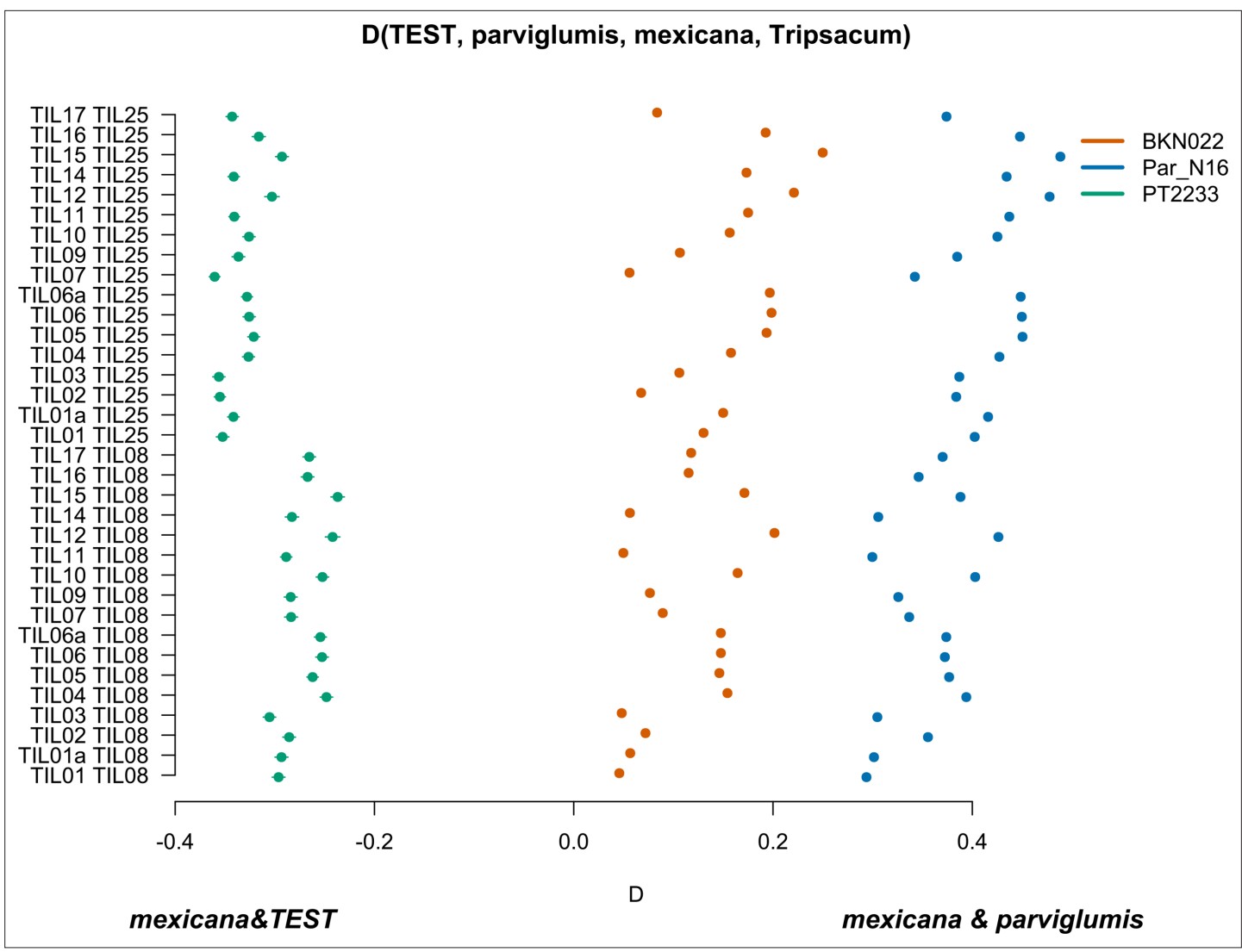

**Figure 4.** Characterization of *mexicana* gene flow with Par_N16 and Mesoamerican landraces. D-statistics were calculated in the form D(TEST, *parviglumis*, *mexicana*, outgroup) (*Figure 4—figure supplement 4*) by comparing 121,842 variant sites shared between Par_N16, *Palomero toluqueño* (PT2233), or *Reventador* (BKN022), and the corresponding SNP variants from teosinte *parviglumis* (TIL01-TIL07, TIL09-17) and two teosinte *mexicana* accessions (TIL25, TIL08). The graph shows the total number of pairwise comparisons (n=34) yielding a negative D for *mexicana* introgression over a test sample or positive D for a higher introgression of *mexicana* and *parviglumis* (*Figure 4—source data 1*). Lines in each dot reflect the standard deviation calculated from 100 jackknife replicates.

The online version of this article includes the following source data and figure supplement(s) for figure 4:

**Source data 1.** D-statistic values for each pairwise combination of (TEST, *parviglumis*, *mexicana*, outgroup).

**Figure supplement 1.** Conceptual representation of the hypothesis testing in the form D(*parviglumis*, *mexicana*, TEST, Tripsacum).

**Figure supplement 2.** Scatterplot of pairwise *mexicana* and *parviglumis* computations of Par_N16, BKN022, and PT2233 D-statistics in the form D(*parviglumis*, *mexicana*, TEST, outgroup).

**Figure supplement 2—source data 1.** D-statistic values for each pairwise combination of (*parviglumis*, *mexicana*, TEST, outgroup).

**Figure supplement 3.** Scatterplot of pairwise *mexicana* and *parviglumis* computations of Par_N16, BKN022, and PT2233 D-statistics in the form D(*parviglumis*, *mexicana*, TEST, outgroup) in which only transversions are included.

**Figure supplement 3—source data 1.** D-statistic values for each pairwise combination of (*parviglumis*, *mexicana*, TEST, outgroup) including only transversions.

**Figure supplement 4.** Conceptual representation of the hypothesis testing in the form D(TEST, *parviglumis*, *mexicana*, Tripsacum).

**Figure supplement 5.** Scatterplot of pairwise *mexicana* and *parviglumis* computations of Par_N16, BKN022, and PT2233 D-statistics in the form D(TEST, *parviglumis*, *mexicana*, outgroup) in which only transversions are included.

*Figure 4 continued on next page*

Figure 4 continued

**Figure supplement 5—source data 1.** D-statistic values for each pairwise combination of (TEST, *parviglumis*, *mexicana*, outgroup) including only transversions.

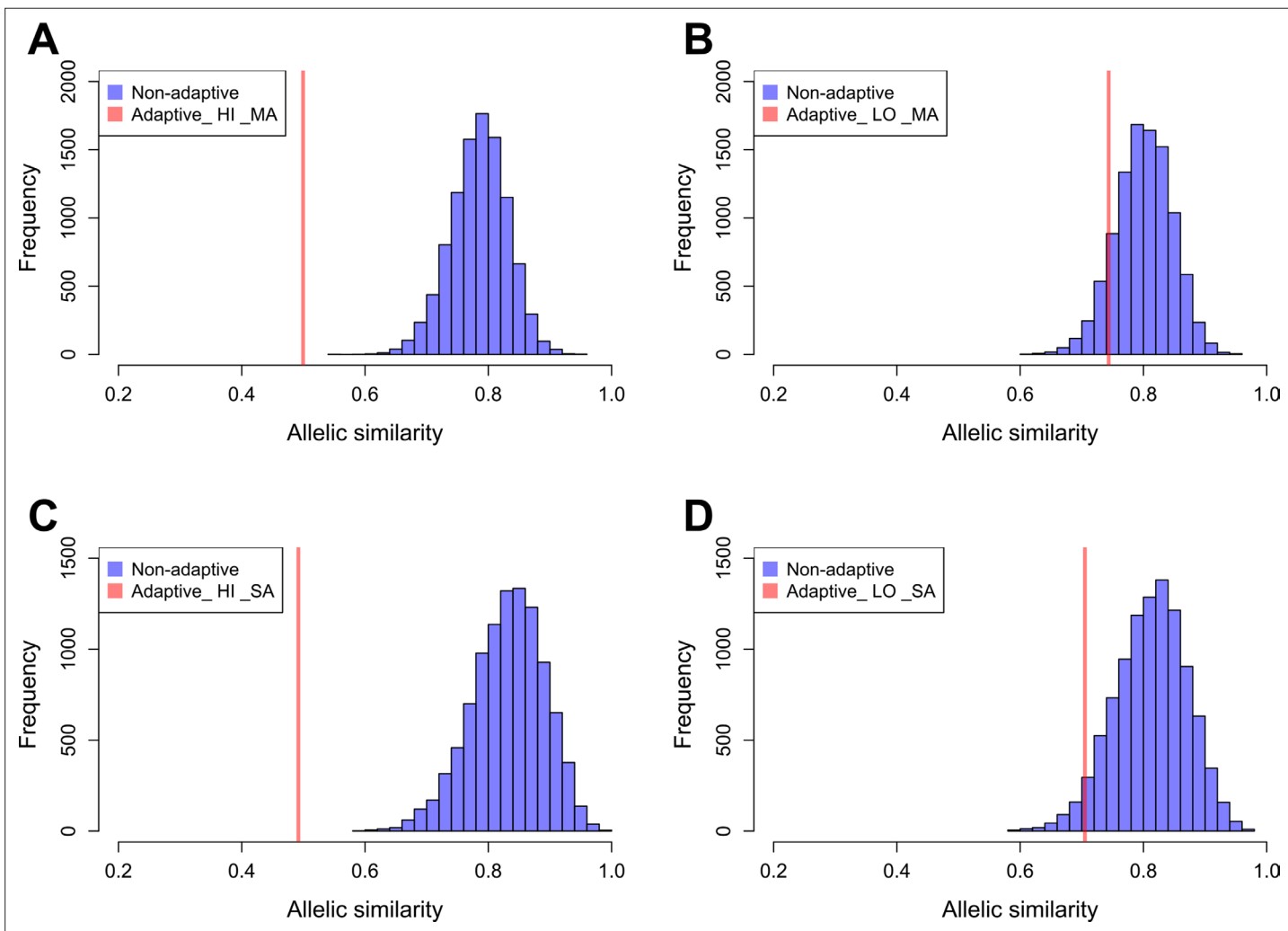

**Figure 5.** Allelic similarities between Par_N16 and landraces from Mesoamerica (MA) and South America (SA). Comparisons involved genome-wide non-adaptive SNPs (blue distributions) and SNPs with significant $F_{ST}$ implicated as adaptive (red lines) at intersected sites between Par_N16 and the reference dataset (*Takuno et al., 2015*). HI, highlands; LO, lowlands; MA, Mesoamerica; SA, South America. In the Y axis is the count of the random samples showing a given allelic similarity; in the X axis is the allelic similarity between Par_N16 and the test population in the intersected sites. (**A**), the mean genome-wide allelic similarity between Par_N16 and highland MA landraces in non-adaptive SNPs is 0.785; the corresponding allelic similarity in adaptive SNPs is 0.4995. (**B**), the mean genome-wide allelic similarity between Par_N16 and lowland MA landraces in non-adaptive SNPs is 0.8; the corresponding allelic similarity in adaptive SNPs is 0.7436. (**C**), the mean genome-wide allelic similarity between Par_N16 and highland SA landraces in non-adaptive SNPs is 0.831; the corresponding allelic similarity in adaptive SNPs is 0.4918. (**D**), the mean genome-wide allelic similarity between Par_N16 and lowland SA landraces in non-adaptive SNPs is 0.812; the corresponding allelic similarity in adaptive SNPs is 0.705 (*Figure 5—source data 1*).

The online version of this article includes the following source data and figure supplement(s) for figure 5:

**Source data 1.** Source data for *Figure 5A–D*.

**Figure supplement 1.** Distribution of intersected SNPs between Par_N16 and landraces from Mesoamerica (MA) and South America (SA).

**Figure supplement 1—source data 1.** Distribution of intersected SNPs between Par_N16 and landraces from Mesoamerica (MA) and South America (SA).

**Figure supplement 2.** Distribution of distances between SNPs in ten random samples of background SNPs.

genome-wide level, Par_N16 is genetically more similar to South American landraces, particularly from the highlands, than to Mesoamerican populations. Meanwhile, at adaptive loci Par_N16 presents less similarity to highlands than to lowlands for both Mesoamerica and South America (*Figure 5*). In these loci, for Mesoamerica, Par_N16 has an average similarity with highland genotypes of 0.4995 and 0.7436 with lowland individuals, while for South America the similarity with highland genotypes was 0.4918, compared to 0.705 for the lowlands (*Figure 5*). Moreover, Par_N16 is significantly less similar in adaptive SNPs with the highland populations from both regions relative to genome-wide expectations (p<0.0001 in both cases); however, its adaptive similarity with lowland populations was significantly reduced for South America (p=0.0386) but not for Mesoamerica (p=0.1116). Nevertheless, allelic similarity for both lowland populations is not far outside of genome-wide expectations, in contrast with highland populations (*Figure 5*). Thus, although Par_N16 is still more adapted to lowland Mesoamerica, it was in the process of adapting to lowland South America.

## Discussion

Paredones ancient maize represents the earliest macro-specimens of maize known to date and was found in Peru 3800 km away from the center of origin. Paredones samples are morphologically similar to extant maize while the earliest maize from Mexico still retained shared morphology and haplotypic diversity with wild populations. Therefore, the recovery of genomic sequences of these early South American populations brings a unique opportunity to reconstruct the adaptation and dispersal processes of maize. All three samples analyzed here are located within the monophyletic clade of maize, indicating that the ultimate origin of ancient Paredones maize is not different from all Mexican landraces examined to date and supporting a single domestication event. The ancient samples were grouped in all cases within a subclade of South American landraces, particularly with the Peruvian landrace Chullpi, suggesting that ancient Paredones maize was already domesticated by 6775–5,324 calibrated BP (at 95.4% probability) and at least partially ancestral to extant South American landraces. A differential *parviglumis* ancestry was observed surrounding domestication loci in previous genomic analyses from ancient and modern maize from South America, and this was interpreted as evidence of stratified domestication, in which one of several partially domesticated lineages arrived early (at least by 7000 BP) in South America and locally evolved all domestication traits (*Kistler et al., 2020*; *Kistler et al., 2018*). An ancient sample located in an ancestral or sister position to the Mesoamerican maize clade would provide evidence to support this model. The phylogenetic position of Paredones samples does not show this pattern (*Figure 3*) and does not support the stratified model as previously proposed (*Kistler et al., 2020*; *Kistler et al., 2018*), but it is compatible with a sequential model of crop evolution in which domestication is the first stage, followed by an increase in frequency of desirable alleles (stage 2), and the formation of cultivated populations adapted to new environments and local preferences (stage 3; *Meyer and Purugganan, 2013*).

In Mesoamerica, maize adaptation to highlands was marked by the introgression of alleles from the highlands teosinte *mexicana*, and maize lineages adapted to Mesoamerican highlands carry this gene flow signal (*Hufford et al., 2013*; *van Heerwaarden et al., 2011*). Previous research suggested that South American highland maize was independently adapted from local lowland germplasm rather than relying on the same allelic diversity that underlies highland adaptation in Mexico (*Takuno et al., 2015*). Our analyses show that there was no significant gene flow between Par_N16 and *mexicana*. If any, it was significantly lower than the gene flow between *mexicana* and lowland landraces from Mexico such as *Reventador* (BKN022), and also significantly lower than the gene flow from *mexicana* to *parviglumis*. This result suggests that the early Paredones maize populations diverged from Mesoamerica without gene flow from *mexicana* or any highlands maize in Mexico, consistent with the idea that *mexicana* introgression into maize populations occurred more recently (1000 generations ago) (*Calfee et al., 2021*). While modern highland South American germplasm shows evidence of *mexicana* introgression (*Swarts et al., 2017*), Paredones maize does not contain detectable *mexicana* allelic diversity and it is possible that the earliest germplasm that was grown in the Andes did not contain it either. This raises the possibility that there is novel highland adaptive diversity harbored by South American landraces; however, more ancient and modern samples, especially from highland Andean locations, are needed to test this hypothesis. In addition, *mexicana* introgression is pervasive across domesticated maize (*Calfee et al., 2021*; *Swarts et al., 2017*); therefore, Paredones ancient

samples might be useful as minimal or no-introgression controls in future studies assessing *mexicana*-maize gene flow.

At the genome-wide level, Par_N16 is more similar to the lowland than to the highland Mesoamerican population (*Figure 5*). Meanwhile, in South America, the genome-wide similarity is higher than in Mesoamerica but especially in the highlands, which is interesting because Paredones is a lowland site. One possible explanation is that Paredones is likely ancestral to both lowland and highland populations (with the latter derived from local lowland landraces), but that subsequent gene flow from Mesoamerica (*Swarts et al., 2017*; *Takuno et al., 2015*) had a greater impact on lowland populations, erasing part of this ancestry. Understanding the process of highland adaptation will require additional sampling in both highlands and lowlands of the Americas.

Allele similarity at SNPs that showed significant $F_{ST}$ values between the highlands and the lowlands in Mesoamerica and South America (*Takuno et al., 2015*), clearly shows that the Paredones sample has far less similarity with highland rather than lowland populations (p<0.0001), as is consistent with their lowland provenance; but surprisingly, sharing higher proportion of adaptive SNPs with lowland Mesoamerican populations than with lowland South American ones, for which similarity was significantly reduced (p=0.0386). These results suggest that ancient Paredones samples were likely better adapted to their ancestral Mexican lowlands than to their new environment in lowland South America. Par_N16 still shares some similarities to lowland South America in adaptive SNPs (unlike highland), evidencing some level of adaptation to the South American environment. In addition, the deficiency of adaptive alleles in this region can be explained if an important portion of the current adaptive alleles were to arrive with later populations or to increase in frequency in the ancestral population in South America after the time of Paredones. Under this perspective, the age of Par_N16 (5583–5324 calibrated BP at 95.4% probability) suggests that a substantial amount of improvement occurred rapidly and specifically in South American lowlands. Additional sampling and archaeological context inferences on the role of maize in ancestral societies are required to better understand how rapidly maize adapted to the South American environment. On the other hand, highland-adaptive alleles are expected to be deleterious in lowlands (*Takuno et al., 2015*), which could explain their scarcity in a lowland sample. Taken together, our evidence suggests that Paredones ancient maize originated in Mesoamerica and arrived to Paredones through a lowland coastal migration route. It also suggests that Paredones lineage was in stage 3 of the crop evolution model mentioned above (*Meyer and Purugganan, 2013*). In the end, adaptations and improvements occurring in both Mesoamerica and South America can explain the rapid evolution that was responsible for the modern phenotype that Paredones maize presents despite its antiquity.

Overall, our results suggest that, unlike in the highlands (*Ramos-Madrigal et al., 2016*; *Vallebueno-Estrada et al., 2016*), domestication occurred in lowlands Mexico before Paredones lineage arrived in South America following a coastal Pacific corridor of cultural and physical goods from Mesoamerica to Peru. Under this scenario, domestication and improvements in Mesoamerican lowlands, migration from there to Peru, and further processes of local adaptation must have occurred throughout ~2500 years, assuming a teosinte-maize divergence time of 9000 years (*Matsuoka et al., 2002*). During this relatively short period, there was no gene flow between *mexicana* and this maize lineage, but there were processes of specific adaptation to South American lowlands, which required expert management in the face of new environmental and socio-economic pressures. These developments fit well within the unique and advanced socio-cultural and technological transformations of coastal Central Andean societies that began to occur during a period of rapid cultural change between ~7500 and 6500 years ago: population growth and aggregation, permanent agro-maritime settlements along the Pacific shoreline, small farming communities in coastal and highland valleys, adoption of a wide variety of cultigens from the highlands and tropical lowlands, introduction of camelid husbandry, and slightly later between 5500 and 5000 years ago, monumental architecture and public ritual, elaborate art and iconography, and craft production (*Bonavia and Grobman, 2017*; *Dillehay et al., 2022*; *Kaulicke and Dillehay, 1999*; *Lavallee, 2000*; *Reindel and Gorbahn, 2018*; *Shady and Leyva, 2002*). These and other socio-cultural transformations generally took place earlier in the Central Andes, perhaps more so along the desert coast, resulting in the establishment of new social organizations, increased agriculture, and intensive landscape modification. Based on current archaeological evidence, these changes represent a package of social and cultural traits so far undocumented in any other region of the Americas during this period.

## Materials and methods

### Radiocarbon dates on maize macro-remains: Chronology and stratigraphy

The chrono-stratigraphic data for the maize remains were published previously in *Grobman et al., 2012* and in *Dillehay, 2017*; *Bonavia and Grobman, 2017*. These data come from archeological excavations at the Huaca Prieta and Paredones sites, both multicomponent Preceramic-aged localities situated on a remnant Pleistocene terrace overlooking the Pacific Ocean on the north coast of Peru. Huaca Prieta is a large artificial earthen and stone mound approximately 165 m long, 85 m wide, and 32 m high. Paredones is an artificial earthen mound and measures approximately 40 long, 23 m wide, and 6 m high. The macro-maize remains from both sites were excavated in deeply stratified, intact (i.e., undisturbed) cultural floors. Stratigraphic Unit 22 at Paredones is the archeological component with the largest and most diversified amount of maize remains, with the oldest C[14] dated cobs derived from the base of this unit (see *Figure 1*), in a single, discrete and intact floor of ~2 cm in thickness and at 5.5 m in depth from the present-day surface (*Bonavia and Grobman, 2017*).

AMS radiocarbon ages on the maize remains were obtained on a burned husk-shank, a slightly charred husk fragment, and an unburned and burned cob fragments by laboratories at the University of Arizona, Beta Analytic Inc, and the Woods Hole Oceanographic Institute. All dates were calibrated using the SHCal04 Southern Hemisphere Calibration 0–11.0 calibrated kyr BP curve (*McCormac et al., 2004*) and shown as 2σ calibrated years BP at 95.4% probability. Furthermore, the dated remains at both sites are chrono-stratigraphically bracketed by and in agreement with more than 165 AMS and OSL dates from the mound and off-mound contexts at both Huaca Prieta and Paredones (*Bonavia and Grobman, 2017*). This allowed the project to cross-reference and control multiple, wide, deep, and intact stratigraphic contexts within and across excavation units at sites, and to document any taphonomic and other potential disturbances that might have affected the integrity of the context and absolute age of the remains. As mentioned above, no taphonomic or other disturbing cultural or geological features were observed in any excavation units that would have altered the integrity and intactness of strata containing the maize remains. Of the 165 total C[14] and OSL assays, only five are anomalous, which is expected for a large batch of assays from different spatial and temporal contexts. In addition, these five anomalous assays are associated with a single unburned plant fragment. All other assays conform to their expected chronological order and stratigraphic position; and agree with the few radiocarbon measures obtained earlier by Bird *Bird and Hyslop, 1985* in the 1950s.

All 151+ C[14] dates were measured on single chunks of wood charcoal, animal bone, burned corn remains, charred short-lived plants (e.g. charred chili pepper and avocado seeds), and burned and unburned cotton textiles recovered from intact hearths, shallow, but deeply buried, food pits (2–3 cm thick), and human burials embedded in discrete floors. All wood charcoal dates were derived from short-lived bushes and trees. No radiocarbon samples were taken from fills, middens, and marine shells. Given the different organic materials dated by four different laboratories over a period of six decades (including Bird's dates by Libby's laboratory in the 1950s), all non-anomalous dates generally agree and overlap chronologically and stratigraphically (at 95.4% probability) for floor sequences in stratigraphic units.

The most complete stratigraphic sequence on dated macro-remains is a series of AMS assays obtained from hearths and shallow features in intact floors from excavation Units 20 and 22 at Paredones. (Note the intact stratigraphy of thin floors in Unit 22 that contains most of the dated specimens in *Figure 1—figure supplement 1*). A charred cob fragment from Floor 6 at a depth of 1.2 m was dated at 4821–4527 cal BP (at the 95.4% probability, AA86934). Dates on single chunks of wood charcoal from hearths embedded in Floors 10, 15, and 16 at depths of 3.8 m, 4.7 m, and 4.9 m, respectively, were processed at 5435–5044 cal BP (at the 95.4% probability, Beta263320), 5585–5325 cal BP (at the 95.4% probability, Beta263321), and 5711–5335 cal BP (at the 95.4% probability, AA86947). Recently, a charred cob fragment from Floor 11 was assayed at 5404–4937 cal BP (at the 95.4% probability, D-AMS 044318), which agrees with its chrono-stratigraphic position and other AMS dates in the site. An articulated maize charred husk and shank fragment from Floor 18 (Par_N1), at a depth of 5.5 m in Unit 22, yielded a date of 6775–6504 cal BP (at the 95.4% probability, OS86020). Underlying this date at a depth of ~5.6 m is an assay of 6640–6319 cal BP (at the 95.4% probability, AA83260) on wood charcoal from a hearth in Floor 24, Unit 22. Two other specimens (Par_N9 and Par_N16) were found in excavation Unit 20, with Par_N9 radiocarbon assayed at 5582–5321 cal BP (at 95.4%

probability, AA86932) and Par_N16 stratigraphically dated at 5603–5333 cal BP (at the 95.4% probability, AA86937) by direct association with wood charcoal in the hearth containing the unassayed unburned cob and assayed charcoal; these dates are in agreement with others taken on burned organic materials, including other burned maize specimens, from levels above and below Par_N1, Par_N9 and Par_N16 (*Bonavia and Grobman, 2017*; *Dillehay et al., 2012*; *Grobman et al., 2012*).

One dated unburned cob segment (Par_N1, OS86020) was problematic. The cob was half burned and half unburned. For comparative purposes, we dated both the unburned and burned segments. The burned segment yielded the date of 6775–6504 cal BP (at the 95.4% probability, OS86020). Four assays were processed on the adjoining uncharred segment, which resulted in anomalous dates that were younger, including one post-bomb date (600–700 cal BP [at the 95.4% probability]). These results indicate that the harder, more durable burned cob Par_N1 segment yielded a reliable C$^{14}$ measure that completely agreed with its stratigraphic position below assays in Floors 10, 15, and 16 and above an assay in Floor 24. It also indicates that assays on the unburned, soft tissue of the cob are anomalous. In general, as discussed below, we discovered that unburned, soft tissue plant remains often produce anomalous assays. In sum, the dates on wood charcoal and burned cobs from Unit 22 are in complete stratigraphic agreement, showing that the anomalous assays on the one unburned segment of cob Par_N1 in Floor 18 are in error.

In regard to the anomalous dates from the unburned segment of cob Par_N1, there is no taphonomic or stratigraphic evidence (e.g. pits, animal burrows, tree roots, postholes, truncated strata) to indicate intrusiveness of younger materials or post-occupation disturbance in any excavated contexts at Paredones (see *Figure 1—figure supplement 1* for the discrete, undisturbed floors in Unit 22 at Paredones). Moreover, all strike and dip measurements taken on piece-plotted artifacts and features embedded in all floors show no upward or downward tilting indicative of post-depositional disturbance, that is, all stone tools, bone, shell, and plant remains and other ecological and artifactual debris laid flat in or on floors, also indicating undisturbed and unimpeachable contexts. All associated features (e.g. pits, food stains) also revealed no disturbance.

The unburned, soft-tissue cob segment with the younger dates may have been contaminated by mold, fungus, and heavy salt saturation from the local seashore environment. We performed SEM analyses of the microscopic cellular structure of unburned and burned cobs and of wood charcoal. Fungal activity was identified only in the soft cellular structure of unburned cobs (*Bonavia, 2011*). No fungal activity was observed in the cells of wood charcoal and of the burned cobs and unburned and burned harder husk and shank fragments. With regard to possible contamination effects from fungi, Darden Hood of Beta Analytic, Inc (*Hood, 2011*) analyzed the unburned cobs and their anomalous dates and proposed reasonable causes to consider: "...uncharred corn acts like a sponge and its integrity is too weak to withstand the [laboratory] pretreatments prior to removing organic contaminants….. the radiocarbon (RC) pretreatments are dissolving the sample just as fast as contamination, resulting simply in a reduction in sample size rather than de-contamination...the corn was being preferentially removed with the [pretreatment] alkali, thereby increasing the concentration of the fungus, the date would come out younger with a higher fungus to corn ratio. As you go deeper [stratigraphically in the site], the corn is more weathered, and more susceptible to removal with the alkali... whereas the fungus is fresh-and-resistant."

Certain conclusions can be drawn from the anomalous dates on the one unburned cob fragment. There is no evidence of post-depositional disturbance at any of the excavated sites. There is no possibility that the younger assays are intrusive. The reliable dates are on hard husks and burned cobs, which have a more rigid, impenetrable, fungi-free cellular structure. This suggests the probability that the other, more porous tissues of soft, uncharred cobs can absorb or allow the growth of some contaminating substances that do not affect the harder tissues of the charred tissue and husks of the maize. In short, the reliable assay for the burned cob segment Par_N1 is 6775–6504 cal BP (at the 95.4% probability, OS86020).

Other dates on burned cob fragments from other localities in and around Paredones and Huaca Prieta were in chrono-stratigraphic order and dated 4149–3839 cal BP (Beta278050), 3956–3704 cal BP (AA86941), 3982–3728 cal BP (AA86931), and 4235–3928 cal BP (AA86946), all at 95.4% probability.

As noted earlier, in 2020, Dillehay and project geologists Steven Goodbred and Elizabeth Chamberlain carried out excavations in a Preceramic domestic site (S-18) located ~3.2 km north of Huaca Prieta. Preceramic corn remains were encountered in floors from the upper to lower intact cultural

layers of the site. As with the Paredones and Huaca Prieta sites, the lower strata contained cobs of the smaller and earliest type of identified corn species, Proto Confite Morocho (*Grobman et al., 2012*). The middle to upper strata yielded the later and slightly larger Preceramic varieties of Confite Chavinese and Proto Alazan. An OSL date from a discrete and intact lower floor containing a hearth with two unburned cob fragments of the Proto Confite Morocho variety assayed at ~7000 +/- 630 years ago or 5610–4350 BCE (*Chamberlain, 2019*). Wood charcoal from the hearth processed at 7162–6914 cal BP (at 95.4% probability, AA75398), thus by direct association dating the two unburned cobs ~7000 years ago.

In South America, maize micro remains (e.g. starch grains, pollen, phytoliths) have been dated 7200–7000 cal BP (*Piperno, 2006*; *Piperno and Pearsall, 1998*; *Zarrillo et al., 2008*) at sites in southwest coastal Ecuador, located ~600 km north of Huaca Prieta and Paredones, and in other localities across the continent ~6500 cal BP and later (*Kistler et al., 2018*).

## Extraction and sequencing of ancient samples

Permits for excavation, analysis, and samples exportation were granted by the Ministerio de Cultura from Peru (Resolución Directorial Nacional No. 414/INC, 2007; Resolución Directorial Nacional N° 000194–2021-DCIA/MC; Resolución Directorial Nacional N° 000168–2022-DCIA-LRS/MC).

Sample processing and DNA extraction were performed following all necessary procedures to avoid human-related or cross-sample contamination in a clean Laboratory optimized for paleogenomics, as previously described (*Vallebueno-Estrada et al., 2016*).

All three maize specimens were sampled using forceps and sterile scalpels. DNA extraction was of 14.2–15 mg of the inner parenchyma tissues at the ancient DNA facilities of UGA Langebio CINVESTAV. Isolation of DNA was carried out in clean laboratory facilities at UGA Langebio and Tuebingen University, Germany, with dedicated reagents and equipment that are frequently sterilized, and UV treated. To prevent cross-sample and human-related contamination, we used new disposable plastic material and filtered pipette tips, and personnel wear protective gear such as full bodysuits, masks, and doubled gloves. Work was conducted in laminar flow hoods. Samples were ground with a mortar and pestle. DNA extraction was conducted using a freshly prepared PTB lysis buffer (PTB 2.5 mM, DTT 50 mM, Proteinase K 0.4 mg/ml, 1% SDS, 10 mM Tris, 10 mM EDTA, 5 mM NaCl) and purified using QIAgen DNEasy Plant Mini kit columns following an established protocol (*Swarts et al., 2017*; *Yoshida et al., 2013*). All shotgun libraries were constructed from 20 µl of ancient maize DNA following a published protocol tested in maize (*Meyer and Kircher, 2010*; *Swarts et al., 2017*) with modifications as suggested in *Meyer et al., 2012*. Libraries were amplified for 10 cycles with unique combinations of two indexing primers (*Kircher et al., 2012*). The quality of libraries was tested using Qubit 2.0 fluorometer (Thermo Fisher) and a High Sensitivity DNA Assay Chip Kit (Agilent, Waldborn Germany) on a Bioanalyzer 2100 (Agilent Technologies).

Non-UDG double index DNA Illumina libraries for each sample were built at Max Planck Institute Tuebingen, using established methodologies for ancient DNA, for subsequent shotgun sequencing. Illumina libraries were sequenced in three different rounds for a total of 83.88 Gb (Par_N1 35.7 Gb, Par_N9 23.48 Gb, Par_N16 24.7 Gb) using Nextseq at Unidad de Genómica Avanzada, Laboratorio Nacional de Genómica para la Biodiversidad, Cinvestav Irapuato.

To increase the endogenous content of the Par_N16 sample a secondary library was sequenced using Nextseq yielding 21.59 of additional Gb. For this library, which originated from a split of the original library already amplified, a final size-selection step was performed using a Pippin-Prep procedure in a 2% agarose cassette (Sage science, Beverly MA) to retain DNA fragments from 150 to 205 bp in size.

## Read processing, mapping, and genotyping

Double Index sequences of 8 nucleotides were used to tag libraries described above. Only reads with the correct index combination were used in downstream analysis. All libraries were filtered to remove adaptors and low-quality reads using Cutadapt (V1.13) (*Martin, 2011*) and keeping reads longer than 30 bp with a quality above 10 Phred score. Repetitive adenines (A) and thymines (T) were invariably removed from read ends.

Filtered reads were mapped against *Z. mays* B73 RefGen_v3 (*Schnable et al., 2009*) using the Burrows-Wheeler analysis (BWA) MEM algorithm with default conditions (*Li and Durbin, 2010*). Reads

with multiple hits were removed using SAMtools map quality filters. As a clonal removal strategy, sequence duplication in reads was filtered with the rmdup function of SAMtools (V1.5) (*Li et al., 2009*), and sequences were locally re-aligned around insertion/deletions (indels) using GATK IndelRealigner (V3.7) (*McKenna et al., 2010*). Mapping efficiency of Paredones ancient samples was 0.1409% for Par_N1 and 0.1612% for Par_N9 and 0.728% for Par_N16. SNP and genotype calling were performed as previously described (*Schubert et al., 2014*; *Seguin-Orlando et al., 2014*). Rescaling of Phred quality scores to account for molecular damage was implemented by using the `--rescale` parameter in mapDamage(V2.2) (*Jónsson et al., 2013*) and keeping reads longer than 30 bp with a quality above 10 Phred score to eliminate reads with low certainty assignments. Variation information was extracted and called using the mpileup and bcftools functions of SAMtools (*Li et al., 2009*), generating a VCF file containing genotypes. This entire pipeline has been already optimized for analyzing ancient maize samples (*Vallebueno-Estrada et al., 2016*).

## Metagenomic analysis and postmortem damage

Cytosine deamination rates and fragmentation patterns were estimated using mapDamage2.2 (*Jónsson et al., 2013*) based on all reads mapping to the B73 reference genome, revealing expected patterns of postmortem damage in the form of C>T substitutions at the 5' termini, and G>A substitutions at the 3' termini. The excess of purines observed near read termini supports fragmentation driven by depurination (*Figure 2*). All indels and sites behaving as molecular damage (CG->TA) (*Gilbert et al., 2003*; *Hofreiter et al., 2001*) were excluded (molecular damage filter). However, heterozygous sites with one variant compatible with molecular damage were transformed into homozygous sites for the variant without damage pattern. A metagenomic filter was applied to discard reads that aligned to sequences in the GenBank National Center for Biotechnology Information database of all bacterial and fungal genomes using default mapping quality parameters of BWA (*Li and Durbin, 2010*). Parallel analyses were conducted using only transversions to assess potential bias introduced by the molecular damage filter.

## Evolutionary analysis and SNP genotype comparisons

Patterns of divergence were analyzed by generating maximum likelihood (ML) trees using Treemix (V1.12) (*Pickrell and Pritchard, 2012*) and the intersection of SNPs passing quality filters for the ancient specimens and 44 selected individuals of the publicly available database HapMap3 without imputation (*Bukowski et al., 2018*). The list of selected individuals is presented in the *Supplementary file 1*. The topologies were generated with each ancient sample individually or including both samples together. In each case, 10,000 bootstrap pseudo-replicates were generated with a parallelized version of a public script (https://github.com/mgharvey/misc_python/blob/master/bin/TreeMix/treemix_tree_with_bootstraps.py; RRID:SCR_023426; *Harvey, 2013*), which uses the sumtree function in DendroPy (*Sukumaran and Holder, 2010*) to obtain a consensus ML bootstrapped tree. The same SNP alignments were also used to assign the identity of each ancient SNP genotype to shared or exclusive SNP genotypes of the selected HapMap3 individuals. According to their SNP identity, the ancient genotypes were assigned exclusively to one of six categories: B73 genotypes, maize landraces genotypes, *Zea mays* ssp. *parviglumis*, *Zea mays* ssp. *mexicana*, *Tripsacum dactyloides*, or those not present in the dataset (ancient sample's private SNPs). Tree topologies were generated based on an intersection between maize HapMap3 concatenated for each genotype. The resulting trees were visualized using figtree software (https://github.com/rambaut/figtree/; RRID:SCR_008515).

## Introgression analysis

Quantification of *mexicana* introgression was performed by D-statistics D (P1, P2, P3, O) calculated from an ABBA(*x*) BABA(*y*) scheme $D=(x-y)/(x+y)$; being *x* the total amount of haplotypes shared between P2 and P3, and *y* the total amount of haplotypes shared between P1 and P3. We used the CalcD function of the evobiR tools package (*Blackmon and Adams, 2015a*; *Blackmon and Adams, 2015b*), performing 100 jackknife replicates using the parameters 'sig.test=J' and 'replicate = 100' (see https://github.com/coleoguy/evobir). Only sites covered in Par_N16 were considered in the controls. To test genetic similarity to highland and lowlands populations, we used GBS public data of South American landraces (*Takuno et al., 2015*). To test *mexicana* introgression, we used all genotype calls intersected between test samples and HapMap3 (*Bukowski et al., 2018*). Two D

forms were tested: (i) D(*parviglumis*, *mexicana*, TEST, outgroup) (*Figure 4—figure supplement 1*) that allows a direct estimation of gene flow of derived alleles back from *parviglumis* or *mexicana* into the test sample, and (ii) D(TEST, *parviglumis*, *mexicana*, outgroup) (*Figure 4—figure supplement 4*) that measures an excess of gene flow between *mexicana* and test compared to the gene flow observed in *mexicana* and *parviglumis*. In both cases *Tripsacum dactyloides* (TDD39103) was set as an outgroup. We used a Wilcoxon nonparametric test for testing differences between positive and negative values using R package 3.6.2 (https://www.rdocumentation.org/packages/stats/versions/3.6.2/topics/wilcox.test). Two sample Kolmogorov-Smirnov (K-S) tests were conducted using ks.test R package version 1.2. (https://www.rdocumentation.org/packages/dgof/versions/1.2/topics/ks.test).

## Coverage at adaptive sites

The reference (published) data consisted of 668 SNPs specific to Mesoamerica and 390 SNPs specific to South America with significant $F_{ST}$ values between highland and lowland populations and therefore considered to be adaptive, as well as 647,821 non-adaptive SNPs (without significant $F_{ST}$ values) from the analyzed panel (*Takuno et al., 2015*). Covered SNPs in Par_N16 were identified by the intersection with the above-mentioned adaptive and non-adaptive (background) SNPs.

To assess the significance of the coverage of Par_N16 in adaptive SNPs, we compared the observed values with the expectations of coverage in non-adaptive SNPs in 10,000 random samples of 668 or 390 SNPs. For Mesoamerica and South America, we generated 10,000 lists, each one comprising 668 or 390 SNPs that were randomly sampled from 63,271 non-adaptive SNPs intersected between Par_N16 and the public dataset (*Takuno et al., 2015*). We recorded the number of SNPs from each of the 668 or 390 lists that were covered in Par_N16, obtaining the respective null coverage distributions representing the coverage expectations. The probability of underrepresentation is then the proportion of the null distribution (n=10,000) that showed the same value or less than that observed in Par_N16 for the 668 or 390 adaptive SNPs.

## Adaptation to Mesoamerican and South American lands

The allelic similarity between Par_N16 and the test population was estimated as

$$S = \frac{1}{N}\sum_{i=1}^{n} f_i$$

in which $S$ is the allelic similarity, $f_i$ is the frequency of the Par_N16 allele in the intersected site $i$ and $N$ is the total number of intersected sites (32 for Mesoamerica and 20 for South America). The values can go from 0 (no similarity) to 1 (complete identity). We estimated allelic similarity between Par_N16 and each of the four test populations (highland Mesoamerica, highland South America, lowland Mesoamerica and lowland South America) at the intersected sites, both adaptive and non-adaptive SNPs.

To assess significance, we generated null distributions of genome-wide similarity expectations for each of the four test populations. We generated 10,000 random samples from the 63,271 non-adaptive SNPs covered in Par_N16, obtaining the allelic similarity for each sample and generating a distribution of similarities for each test population (*Supplementary file 3A and B*). Each random sample contained the same number of SNPs as the number of adaptive SNPs covered in Par_N16 (32 for Mesoamerica and 20 for South America). The statistical significance of the reduction in adaptive similarity relative to genome-wide similarity can be estimated as the proportion of the null distribution (n=10,000) that show the same or less similarity than that observed in Par_N16 for the 32 or 20 covered adaptive SNPs. The mean genome-wide similarity is the average of 10,000 similarity values obtained from the corresponding random samples.

## Acknowledgements

We thank Hilda Ramos-Aboites and Christian Martinez-Guerrero for technical support, as well as Peruvian archeologists and the Ministry of Culture in Lima, Peru, for granting permission to carry out the archeological research and analysis. We thank Hernán A Burbano for advice on molecular biology protocols and for hosting MV-E. during his visit to Germany. The GBS data from South American landraces *Takuno et al., 2015* used in the adaptation analysis were kindly provided by Jeffrey Ross-Ibarra. MV-E, GGH-R, EG-O. and IL-V were recipients of a graduate scholarship from Consejo Nacional

de Ciencia y Tecnología (CONACyT). This research was supported by a CONACyT grant to JPV-C. (CB256826), the INAH-Cinvestav Tehuacan initiative, and Cinvestav internal funds assigned to JPV-C. and to RM.

## Additional information

### Funding

| Funder | Grant reference number | Author |
|---|---|---|
| Consejo Nacional de Ciencia y Tecnología | Graduate Student Fellowship | Miguel Vallebueno-Estrada |
| Consejo Nacional de Ciencia y Tecnología | CB256826 | Jean-Philippe Vielle-Calzada |
| INAH-Cinvestav | Tehuacan initiative | Jean-Philippe Vielle-Calzada |
| Centro de Investigación y de Estudios Avanzados del Instituto Politécnico Nacional | Internal funds | Jean-Philippe Vielle-Calzada Rafael Montiel |

The funders had no role in study design, data collection and interpretation, or the decision to submit the work for publication.

### Author contributions

Miguel Vallebueno-Estrada, Conceptualization, Data curation, Software, Formal analysis, Validation, Investigation, Visualization, Methodology, Writing – original draft, Writing – review and editing; Guillermo G Hernández-Robles, Data curation, Formal analysis; Eduardo González-Orozco, Ivan Lopez-Valdivia, Teresa Rosales Tham, Víctor Vásquez Sánchez, Formal analysis; Kelly Swarts, Methodology, Writing – review and editing; Tom D Dillehay, Conceptualization, Formal analysis, Writing – original draft, Writing – review and editing; Jean-Philippe Vielle-Calzada, Conceptualization, Formal analysis, Funding acquisition, Investigation, Methodology, Resources, Supervision, Writing – original draft, Writing – review and editing; Rafael Montiel, Conceptualization, Resources, Formal analysis, Supervision, Investigation, Methodology, Writing – original draft, Writing – review and editing

### Author ORCIDs

Miguel Vallebueno-Estrada (ID) http://orcid.org/0000-0001-8647-8758
Teresa Rosales Tham (ID) http://orcid.org/0000-0003-2555-6032
Víctor Vásquez Sánchez (ID) http://orcid.org/0000-0003-4777-9237
Jean-Philippe Vielle-Calzada (ID) http://orcid.org/0000-0002-3017-2490
Rafael Montiel (ID) http://orcid.org/0000-0001-8052-0679

### Decision letter and Author response

Decision letter https://doi.org/10.7554/eLife.83149.sa1
Author response https://doi.org/10.7554/eLife.83149.sa2

## Additional files

### Supplementary files

• Supplementary file 1. Description of all accessions included in the phylogenetic analysis.

• Supplementary file 2. Median distances between the 20 South American adaptive SNPs (left) and the 32 Mesoamerican adaptive SNPs (right) covered in Par_N16.

• Supplementary file 3. (**A**) Allele frequency of highland and lowland Mesoamerican landraces at sites intersected with Par_N16 from 668 adaptive SNPs (2A). (**B**) Allele frequency of highland and lowland South American landraces at sites intersected with Par_N16 from 390 adaptive SNPs.

• MDAR checklist

## Data availability

Sequence data generated for this study have been deposited in the European Nucleotide Archive, Bioproject PRJEB61159 https://www.ebi.ac.uk/ena/browser/view/PRJEB61159 (accession no. ERS13471621 for Par_N1; ERS13471622 for Par_N9; and ERS13471623 for Par_N16). Other newly created datasets are provided in the supplemental material (*Figure 5—source data 1*).

The following datasets were generated:

| Author(s) | Year | Dataset title | Dataset URL | Database and Identifier |
|---|---|---|---|---|
| Vallebueno-Estrada M, Hernández-Robles GG, González-Orozco E, Lopez-Valdivia I, Rosales Tham T, Sánchez Vásquez V, Swarts K, Dillehay T, Vielle-Calzada JP, Montiel R | 2023 | Ancient maize samples from Paredones, Peru | https://www.ncbi.nlm.nih.gov/biosample/?term=SAMEA111374635 | NCBI BioSample, SAMEA111374635 |
| Vallebueno-Estrada M, Hernández-Robles GG, González-Orozco E, Lopez-Valdivia I, Rosales Tham T, Sánchez Vásquez V, Swarts K, Dillehay T, Vielle-Calzada JP, Montiel R | 2023 | Ancient maize samples from Paredones, Peru | https://www.ncbi.nlm.nih.gov/biosample/?term=SAMEA111374634 | NCBI BioSample, SAMEA111374634 |
| Vallebueno-Estrada M, Hernández-Robles GG, González-Orozco E, Lopez-Valdivia I, Rosales Tham T, Sánchez Vásquez V, Swarts K, Dillehay T, Vielle-Calzada JP, Montiel R | 2023 | Ancient maize samples from Paredones, Peru | https://www.ncbi.nlm.nih.gov/biosample/?term=SAMEA111374633 | NCBI BioSample, SAMEA111374633 |
| Vallebueno-Estrada M, Hernández-Robles GG, González-Orozco E, Lopez-Valdivia I, Rosales Tham T, Sánchez Vásquez V, Swarts K, Dillehay T, Vielle-Calzada JP, Montiel R | 2023 | Domestication and lowland adaptation of coastal preceramic maize from Paredones, Peru | https://www.ncbi.nlm.nih.gov/bioproject/PRJEB61159 | NCBI BioProject, PRJEB61159 |
| Vallebueno-Estrada M, Hernández-Robles GG, González-Orozco E, Lopez-Valdivia I, Rosales Tham T, Sánchez Vásquez V, Swarts K, Dillehay T, Vielle-Calzada JP, Montiel R | 2023 | Domestication and lowland adaptation of coastal preceramic maize from Paredones, Peru | https://www.ebi.ac.uk/ena/browser/view/ERS13471621 | European Nucleotide Archive, ERS13471621 |
| Vallebueno-Estrada M, Hernández-Robles GG, González-Orozco E, Lopez-Valdivia I, Rosales Tham T, Sánchez Vásquez V, Swarts K, Dillehay T, Vielle-Calzada JP, Montiel R | 2023 | Domestication and lowland adaptation of coastal preceramic maize from Paredones, Peru | https://www.ebi.ac.uk/ena/browser/view/ERS13471622 | European Nucleotide Archive, ERS13471622 |

*Continued on next page*

*Continued*

| Author(s) | Year | Dataset title | Dataset URL | Database and Identifier |
|---|---|---|---|---|
| Vallebueno-Estrada M, Hernández-Robles GG, González-Orozco E, Lopez-Valdivia I, Rosales Tham T, Sánchez Vásquez V, Swarts K, Dillehay T, Vielle-Calzada JP, Montiel R | 2023 | Domestication and lowland adaptation of coastal preceramic maize from Paredones, Peru | https://www.ebi.ac.uk/ena/browser/view/ERS13471623 | European Nucleotide Archive, ERS13471623 |

The following previously published dataset was used:

| Author(s) | Year | Dataset title | Dataset URL | Database and Identifier |
|---|---|---|---|---|
| Bukowski R, Guo X, Lu Y, Zou C, He B, Rong Z, Wang B, Xu D, Yang B, Xie C, Fan L, Gao S, Xu X, Zhang G, Li Y, Jiao Y, Doebley JF, Ross-Ibarra J, Lorant A, Buffalo V, Romay MC, Buckler ES, Ware D, Lai J, Sun Q, Xu Y | 2017 | Maize Haplotype Map version 3 | https://www.ncbi.nlm.nih.gov/bioproject/PRJNA399729 | NCBI BioProject, PRJNA399729 |

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
