## [Editor Report]

In this important article, the authors characterize ancient DNA from maize unearthed in archaeological contexts from Paredones and Huaca Prieta in the Chicama river valley of Peru, recovered by painstakingly controlled excavation. The genetic evidence, while from a small number of samples, is compelling, although the dating evidence has to rely on archaeological context, which fortunately is excellent. The difficulties of direct radiocarbon dating of the samples in this case are appropriately discussed by the authors.

---

## [Decision Letter]

**Decision letter after peer review:**

Thank you for submitting your article "Domestication and lowland adaptation of coastal preceramic maize from Paredones, Peru" for consideration by *eLife*. Your article has been reviewed by 2 peer reviewers, and the evaluation has been overseen by a Reviewing Editor and Detlef Weigel as the Senior Editor. The reviewers have opted to remain anonymous.

Essential revisions:

As you can see from the individual reviews appended below, only a limited reanalysis of the sequencing data is asked for. The most important request is to confirm the dating of the samples. Please respond to me directly whether or not that will be possible, and in case it will not, how you will address this concern in the revision by textual changes.

*Reviewer #1 (Recommendations for the authors):*

For the comparison of selected SNPs to the background, I think it would help to construct a more appropriate null model such as where the distribution of SNPs in the null is similar to the selected ones so that non-independence among them can be properly represented in the null.

Some writing comments:

The Introduction section is good, but it lacks a pitch for a general audience.

For this statement, "higher levels of genomic diversity than expected (Ramos-Madrigal et al., 2016; Vallebueno-Estrada et al., 2016)" it would help to say what is the expectation.

In the Discussion, it would help to mention what evidence Kistler et al. used for their alternative hypothesis.

*Reviewer #2 (Recommendations for the authors):*

Statements to the effect that "Par_N1 is the most ancient specimen, dating 5,900{plus minus}40 14C years BP (6775-6504 calibrated BP at 95.4% probability), and obtained from archeological Unit 22 (Figure 1C and Figure S1)." does not provide the laboratory number nor additional laboratory information that might help interpret the veracity of this particular radiocarbon date. While you are more forthcoming when discussing the other dated specimens subject to ancient DNA extraction/analysis, to wit: "The other two samples were found in Unit 20, Par_N9 which radiocarbon assayed at 5582-5321 calibrated BP (at 95.4% probability, AA86932), and Par_N16 which stratigraphically dated by direct association with wood charcoal in a hearth at 5603-5333 calibrated BP (at 95.4% probability, AA86937; see Figure 1C and Table 1)" the δ13C value for specimen Par_N9 is -23.5, far outside the expected range for maize. It would appear possible the dates are accurate but the material dated is not the maize cob. Alkaline marine sediments could provoke the exchange of carbon with environmental constituents, alternatively, environmental carbonates could replace organic carbon. Perhaps the laboratory conducting the isotopic analysis would have noticed the disparity and would have evaluated their base/acid/base washes for efficacy. I would suggest the authors redate these three specimens and if they do, I hope that they will prepare the laboratory personnel with depositional environment information so appropriate laboratory protocols can be implemented.

---

## [Author Response]

Reviewer #1 (Recommendations for the authors):For the comparison of selected SNPs to the background, I think it would help to construct a more appropriate null model such as where the distribution of SNPs in the null is similar to the selected ones so that non-independence among them can be properly represented in the null.

The SNPs in Takuno *et al.* (2015) are in general at a median distance of 353 bp from each other. The 20 adaptive sites covered in Par_N16 for South America (SA) are at a median distance of 8,301,843 bp (approximately 8.3 Mbp), while the 32 for Mesoamérica (MA) are at a median distance of 24,295,968 bp (approximately 24.3 Mbp). SNPs in five pairs from Mesoamerica are closer than 100 bp between them, but each pair is at a considerable distance (beyond 1 cM) from each other and from other SNPs covered in Par_N16. This same happens for only one SNP pair from South America. Then, in general, the covered adaptive SNPs are not clustered. For our random samples, the range of genomic distances between SNPs is similar to those of adaptive SNPs. This shows that our null distributions are adequate for our statistical purposes. The genomic positions of covered adaptive sites in Par_N16 are now included in a new Table in the revised version (Supplementary File 2). We have included these observations in the main text (section *Specific adaptation to lowlands in Mesoamerica and South America*), as follows: “In general, adaptive SNPs represented in Par_N16 were not clustered. The 20 South American adaptive SNPs are at a median distance of 8,301,843 bp, while the 32 Mesoamerican SNPs are at a median distance of 24,295,968 bp (Supplementary File 2). SNPs in five pairs from MA are closer than 100 bp between them, but each pair is at a considerable distance (beyond 1 cM) from each other and from other SNPs. This same happens for only one SNP pair from SA. Then, although at low proportions, the adaptive SNPs in Par_N16 are a bona fide representation of different genomic responses to selection pressures…” and “We analyzed some of these random samples and observed a similar behavior as the adaptive SNPs regarding the range of distances between SNPs (Figure 5—figure supplement 2).”

Some writing comments:The Introduction section is good, but it lacks a pitch for a general audience.

We have included the following paragraph at the start of the Introduction to improve this aspect: “Maize constituted 12% of global crop production in 2019, second only to sugar cane (FAO, 2021). Like many crop plants, global maize production is threatened by climate change, especially in the middle to low latitudes (Li et al., 2022) where maize is dominant. Maize has the allelic diversity to adapt, but much of this variation is partitioned differentially across populations (Hufford et al., 2012; Romay et al., 2013; Zila et al., 2013). Understanding the development of population dynamics in maize allows us to not only better understand the evolutionary processes that produced a globally important crop plot but will highlight populations that can be used in breeding to adapt to changing climates.”

For this statement, "higher levels of genomic diversity than expected (Ramos-Madrigal et al., 2016; Vallebueno-Estrada et al., 2016)" it would help to say what is the expectation.

Genetic diversity in maize is considered to be low, due to the domestication bottleneck (Meyer and Purugganan, 2013), especially in domestication loci (Yamasaki *et al.* 2005). Is in this sense that those samples showed more diversity than expected. We improved this sentence in the Introduction of the revised text as follows: “Genomic investigations of archeological samples from the Tehuacan highland site suggested that the dispersal of maize to the highlands of México was complex, as early-arriving maize populations retained higher levels of genomic diversity than expected for domesticated plants (Ramos-Madrigal et al., 2016; Vallebueno-Estrada et al., 2016)."

In the Discussion, it would help to mention what evidence Kistler et al. used for their alternative hypothesis.

We now mention the evidence used by Kistler *et al.*, in the Discussion section of the revised version, as follows: “A differential *parviglumis* ancestry was observed surrounding domestication loci in previous genomic analyses from ancient and modern maize from South America, and this was interpreted as evidence of stratified domestication, in which one of several partially domesticated lineages arrived early (at least 7,000 BP) in South America and locally evolved all domestication traits (Kistler et al., 2020, 2018).”

Reviewer #2 (Recommendations for the authors):

Statements to the effect that "Par_N1 is the most ancient specimen, dating 5,900{plus minus}40 14C years BP (6775-6504 calibrated BP at 95.4% probability), and obtained from archeological Unit 22 (Figure 1C and Figure S1)." does not provide the laboratory number nor additional laboratory information that might help interpret the veracity of this particular radiocarbon date. While you are more forthcoming when discussing the other dated specimens subject to ancient DNA extraction/analysis, to wit: "The other two samples were found in Unit 20, Par_N9 which radiocarbon assayed at 5582-5321 calibrated BP (at 95.4% probability, AA86932), and Par_N16 which stratigraphically dated by direct association with wood charcoal in a hearth at 5603-5333 calibrated BP (at 95.4% probability, AA86937; see Figure 1C and Table 1)" the δ13C value for specimen Par_N9 is -23.5, far outside the expected range for maize. It would appear possible the dates are accurate but the material dated is not the maize cob.

The laboratory number associated with Par_N1 has been added both in the main text and in Table 1. Regarding δ13C value, see the response to point 1 of Reviewer 2 in the public reviews section.

Alkaline marine sediments could provoke the exchange of carbon with environmental constituents, alternatively, environmental carbonates could replace organic carbon. Perhaps the laboratory conducting the isotopic analysis would have noticed the disparity and would have evaluated their base/acid/base washes for efficacy. I would suggest the authors redate these three specimens and if they do, I hope that they will prepare the laboratory personnel with depositional environment information so appropriate laboratory protocols can be implemented.

As discussed in Grobman *et al.* (2012) and Dillehay (2017), the dating procedures and results for the maize remains were discussed and studied with radiocarbon laboratories, resulting in the probability that fungal intrusion off-set the assay of the anomalous unburned cob fragment (see discussion in Grobman *et al.* 2012 and in the main text).

References

Meyer RS, Purugganan MD. Evolution of crop species: genetics of domestication and diversification. Nat Rev Genet. 2013 Dec;14(12):840-52. doi: 10.1038/nrg3605.

Yamasaki M, Tenaillon MI, Bi IV, Schroeder SG, Sanchez-Villeda H, Doebley JF, Gaut BS, McMullen MD. A large-scale screen for artificial selection in maize identifies candidate agronomic loci for domestication and crop improvement. Plant Cell. 2005 Nov;17(11):2859-72. doi: 10.1105/tpc.105.037242.